# A retrospective cohort study evaluating the association between opioid and alcohol-related emergency department presentations and the subsequent risk of hospitalization

Kristen A. Morin [1,2,3,4]*, Laura Hill[1,5] Shannon Knowlan[1], Adele Bodson[1], Paola Nikodem[1], Natalie Aubin[1], David C. Marsh [1,2,3,4], Tara Leary[1,4]

1 Health Science North, Sudbury, Ontario, Canada, 2 ICES North, Sudbury, Ontario, Canada, 3 Centre for Social Accountability, Sudbury, Ontario, Canada, 4 Northern Ontario School of Medicine, Sudbury, Ontario, Canada, 5 Laurentian University, Sudbury, Ontario, Canada

* kmorin@nosm.ca

## Abstract

### Objective

Our objective was to evaluate the association between two types of substance use presentations in the emergency department (ED) (opioid and alcohol) and the subsequent risk of hospital admission.

### Methods

The study is a retrospective observational cohort study using administrative data from all patients presenting with substance use disorder (SUD) at Health Sciences North (HSN) from January 1, 2018, to August 31, 2023. Patients were placed in two groups: those with alcohol-related presentations and those with opioid-related presentations. The outcome was the time and number of ED visits between the index ED visit and first admission to the hospital for the substance-related presentation.

### Results

A total of 5,240 individuals (45.98%) presented with opioid use, and 6,140 individuals (45.61%) presented with alcohol use. The opioid group was younger (mean age = 36.86 years, compared to 44.58 years in the alcohol group) and had higher rates of current homelessness (37.47% vs. 9.63%), a higher prevalence of mental disorders (15.71% vs. 10.68%), and a greater likelihood of being diagnosed with cellulitis (5.24% vs. 0.52%). Despite similarities in 30-day ED revisits (41.53% for alcohol vs. 40.88% for opioids) and mean length of stay (12.16 days for opioids vs. 10.04 days for alcohol), individuals in the opioid group had a higher likelihood of inpatient admission with each additional ED visit (hazard ratio = 1.28, 95% CI [1.19, 1.37]).

**Data availability statement:** Data cannot be shared publicly because of patient confidentiality laws. Data are available from the HSN Ethics Committee (contact via kmorin@nosm.ca) for researchers who meet the criteria for access to confidential data.

**Funding:** TL - Northern Ontario Academic Medicine Assiciation (NOAMA) Grant C-23-14 https://www.noama.ca/ Funders had no role in the study design, data collection, analysis or decision to publish. Nor did they have a role in the preperation of the manuscript.

**Competing interests:** The authors have declared that no competing interests exist.

**Abbreviations:** AMCS, Addiction Medicine Consult Service; AMU, Addiction Medicine Unit; DAD, Drug and Alcohol Dependence; ED, Emergency Department; HR, Hazard Ratio; HSN, Health Sciences North; ICD, International Classification of Disease; KPI, Key Performance Indicators; NACRS, National Ambulatory Care Reporting System; OR, Odds Ratio; SUD, Substance Use Disorder.

## Conclusion

Our findings highlight the healthcare needs of individuals presenting to the ED with opioid use versus alcohol use, with opioid-related cases involving more acute and complex healthcare presentations.

## Introduction

Alcohol and opioid use disorders are the most prevalent, pervasive, and longest-standing substance use disorders (SUDs) in the world [1, 2, 3]. Alcohol and opioid use present significant public health challenges worldwide [1,3]. These substances contribute to morbidity, mortality, and high healthcare utilization across diverse populations [1,3]. The consequences of alcohol and opioid misuse extend beyond individual health, impacting families, communities, and healthcare systems [4,5]. Emergency departments (EDs) play a pivotal role in addressing the acute consequences of substance use disorders, serving as primary points of entry into the healthcare system for individuals in crisis [6, 7, 8, 9].

While much of the recent focus has been on opioids due to the unprecedented rise in opioid-related poisonings and deaths [10,11], longitudinal studies of individuals with substance use disorders have largely neglected the distinct harms associated with alcohol use. Despite its well-documented burden on public health, alcohol use has received comparatively less attention in the context of healthcare utilization and acute care studies. This oversight is significant given that in 2016, more than 4% of all deaths in Canada were attributed to alcohol use, which also contributed to over 6% of all potential years of life lost among individuals aged 15 years and older [12]. While research on opioid-related health service utilization has expanded in recent years, studies examining patterns of healthcare use for alcohol-related presentations, particularly in comparison to opioid-related presentations remain scarce. Addressing this gap is crucial for understanding the unique healthcare trajectories and needs of individuals with alcohol-related disorders, thereby enabling the development of more comprehensive and effective substance use care strategies.

Several reports have focused on identifying the burden of substance use presentations in hospitals, which is used as a key performance indicator (KPI) in Ontario [13]. However, none have modeled the association between types of substance use presentations in the ED (opioid and alcohol) and the subsequent risk of hospital admission. Examining the time and number of ED episodes leading up to a hospital admission can indicate the clinical severity of conditions within a population [14]. Examining clinical severity by way of acute care use patterns can demonstrate the differential impacts of alcohol and opioid use on healthcare systems and shed light on whether the timing of interventions can improve outcomes. This may provide evidence for healthcare providers to tailor resources and interventions accordingly.

Therefore, our objective is to evaluate the association between opioid and alcohol presentations in the ED and the subsequent risk of hospital admission. Previous studies have independently explored hospital admissions associated with alcohol-related

presentations [22] and opioid-related presentations [6,8,21], yet few have directly compared these two groups within the same analytical framework. Comparative investigations examining hospital utilization for these two common SUD presentations remain limited. Incorporating both conditions into a unified analysis can provide critical insights into differential healthcare trajectories and inform targeted resource allocation. This analysis evaluates whether opioid- versus alcohol-related ED presentations are associated with differences in hospitalization risk, including both the time to admission and the number of ED visits preceding admission. It is hypothesized that individuals presenting with opioid-related concerns will exhibit a higher risk of inpatient admission and require more ED visits prior to hospitalization than those presenting with alcohol-related concerns.

## Methods

### Design and setting

This study is a retrospective cohort analysis using administrative data. The study focused on patients who had an index visit to either the ED or hospital facility at Health Sciences North (HSN) in Sudbury, Ontario, Canada. The retrospective data were obtained on March 21, 20223. For the purpose of this study, all identifying information was removed for the analysis. Hospital visits were specifically identified based on substance use being listed as either the primary or secondary reason for their visit, as classified within Chapter 5 of the Canadian version of the International Classification of Diseases, Tenth Revision (ICD-10-CA) [15], under codes F10-19. All data underwent de-identification procedures, and ethical approval was obtained in accordance with local regulations, as reviewed by HSN's Research Ethics Board #23–039. Health Sciences North HSN is an acute care hospital situated in a small urban setting in Northern Ontario, serving approximately 570,000 individuals across Northeastern Ontario. The data utilized in this study originate from administrative sources.

### Data sources and study population

This study was conducted using administrative by leveraging large datasets routinely collected by HSN. These data are routinely gathered for administrative, billing, and regulatory purposes. We received patient-level data and assigned a unique identifierto each patient. All identifying information was removed for analysis and reporting. All study participants were identified between January 1, 2018, and August 31, 2023, with patient outcome accrual ending on September 30th, 2023. Data was extracted on March 21, 2023. All data was anonymized using unique patient identifiers. For this study, we excluded all patients who did not have an indication of opioid or alcohol use in their records. The Discharge Abstract Database (DAD) [15] contains detailed information on all hospital admissions and discharges, and the National Ambulatory Care Reporting System database (NACRS) [16] contains information on hospital ED visits and discharges including ICD-10 [17] diagnosis codes were used as source data for the study.

### Exposure groups

The exposure groups are defined with ICD 10 codes as Alcohol-related ED visit and Opioid-related ED visits (Alcohol F10 and Opioid F11 as primary or secondary diagnosis).

If both alcohol and opioids were present (n = 320, 4.9%), we defaulted the person into the opioid group [13].

### Index event

Index events/cohort entry is defined as the discharge date of the ED for a primary or secondary diagnosis with DAD or NACRS discharge codes of F10-19 within ICD-10-CA Chapter 5) [13].

### Outcomes

The primary outcomes were predetermined and defined as the first hospital admission after the index event occurring within the study period. If no admission event occurred within the study period, the patients were considered right

censored. The secondary outcome studied was 30-day ED revisits defined as all cause visit to ED within 30-days of the index visit. The 30-day window starts when the index visit discharge date occurred. If a revisit does not occur within 30-days of the index date, the 30-day window is re-started upon the presentation which becomes the new index date.

## Covariates

Covariates for the study were collected at the time of admission to ED or hospital and are considered baseline covariates. Age, biological sex, homelessness, and visits to the ED or hospital for mental health, and primary or tertiary occurrence of alcohol or opioid ICD-10 codes [17]. Two criteria were used to identify homelessness for this study: 1) patients were flagged by identifying an ICD-10 code Z59; 2) trained abstractors examine the EMR for physician notes of homelessness, and descriptors of homelessness (no fixed address). Mental health diagnoses were determined using all ICD-10 F codes, excluding F1, which indicates substance use. Two in-hospital intervention groups were adjusted for as well: [1] Addiction Medicine Consult Service (AMCS) which provides substance use support to all patients both hospitalized and in ED; [2] and the Addiction Medicine Unit (AMU), an inpatient medical unit offering stabilization support.

Triage level at the time of ED presentation was also included as a covariate to assess initial acuity and clinical urgency, using the standardized five-level Canadian Triage and Acuity Scale (CTAS), where level 1 represents the most urgent and level 5 the least.

The selected covariates are relevant as they capture baseline patient characteristics and clinical factors that may influence health outcomes, including age, sex, homelessness, mental health status, and substance use history [5,16–18]. Adjusting for these covariates, along with intervention groups and triage level, allows for a more accurate assessment of the impact of treatment and clinical acuity on patient outcomes while accounting for potential confounders.

## Statistical analysis

Descriptive analysis was used to summarize the baseline characteristics of our study population for both continuous and categorical variables [18]. For continuous variables, we reported the mean and standard deviation. Categorical variables were summarized using frequencies and percentages. Logistic regression models were used to determine the association between alcohol and opioid presentations and covariates with 30-day revisits. In addition to logistic regression models, Cox proportional hazards models [20,21] were used to investigate the exposure groups and covariate factors associated with time from initial ED visit to hospital admission and the number of ED visits leading up to hospital admission. All statistical tests were at the p = 0.05 and 95% confidence threshold for statistical significance. Kaplan-Meier curves were fit to measure the raw admission probabilities within 100 days of the initial ED visit, and the number of ED visits leading up to the first admission. Both Kaplan-Meier curves were tested for differences with the Mantel-Haenszel test. All statistical analyses were computed using SAS for academics 3.1.0. Analyses were conducted to evaluate whether opioid- versus alcohol-related ED presentations were associated with differences in the time from ED visit to hospitalization and the number of ED visits prior to hospitalization, in line with the study hypotheses.

## Results

A total of 5,240 (45.98%) patients were in the opioid group, and 6,140 (45.61%) patients were in the alcohol group. We examined differences between these groups in terms of time to hospitalization and number of ED visits prior to admission, to assess our hypothesis that individuals with opioid-related presentations would experience higher hospitalization risk and require more ED visits before admission. There was a statistically significant higher proportion of females in the opioid group (37.93%) compared to the alcohol group (30.18%). The opioid group was significantly younger than the alcohol group, with a mean age of 36.86 years (SD = 11.76) versus 44.58 years (SD = 17.71), respectively. Regarding homelessness, 9.63% of individuals in the alcohol group were homeless compared to 37.47% of those in the opioid group, a significant difference (p < .001). A higher proportion of individuals in the alcohol group resided in Northern Ontario

(91.99%) compared to the opioid group (71.81%). Mental health disorders were identified in 10.68% of individuals in the alcohol group and 15.71% of those in the opioid group (p < .001). Similarly, cellulitis was more common in the opioid group (5.24%) compared to the alcohol group (0.52%), a significant difference (p < .001). Results are presented in Table 1.

In terms of health service use, admissions to the AMU differed significantly, with 4.75% of individuals in the alcohol group and 13.61% of individuals in the opioid group being admitted to the AMU. Similarly, the use of the AMCS was significantly different (p < .001), with 20.90% of the opioid group and 11.18% of the alcohol group receiving services from AMCS. A total of 5.01% of individuals in the alcohol group and 3.69% of individuals in the opioid group were triaged at level 1 (p = .0002). Regarding 30-day ED revisits, there was no significant difference, with 40.88% of individuals in the opioid group and 41.53% of individuals in the alcohol group revisiting the ED within 30 days (p = .6). The mean length of stay was significantly different, with the opioid group having a mean of 12.16 days (SD = 22.76) and the alcohol group having a mean of 10.04 days (SD = 19.39). Additionally, the mean number of ED presentations was 6.40 (SD = 6.39) for the opioid group and 6.21 (SD = 6.21) for the alcohol group. Results are presented in Table 2.

When studying factors associated with 30-day revisits, we found that experiencing homelessness was associated with a significantly higher likelihood of 30-day revisits (OR = 1.40, 95% CI [1.20, 1.65]) compared to non-homeless individuals (reference group). Additionally, individuals with no mental health disorders (reference group) had a significantly lower likelihood of 30-day revisits (OR = 0.80, 95% CI [0.69, 0.93]). Lastly, triage levels 4 and 5 (lowest acuity) were associated with a significantly higher likelihood of 30-day revisits (OR = 1.57, 95% CI [1.11, 2.21] and OR = 2.22, 95% CI [1.34, 3.70], respectively).

The data indicated that the opioid group's likelihood of inpatient admission, in relation to the time from first ED visits to hospitalization, was not significantly higher than that of the alcohol group. Fig 1 describes the cumulative incidence of hospital admission by exposure group (days leading to hospital admission). Among individuals presenting with opioid use, 14% were admitted to the hospital the next day, and 10% were admitted within three days after discharge from the ED. In

**Table 1. Cohort characteristics by exposure groups.**

|  | Alcohol | | Opioids | | p value |
|---|---|---|---|---|---|
|  | n = 6140 | 45.61% | n = 5240 | 45.98% |  |
| Sex n (%) |  |  |  |  | <0.001 |
| male | 4,287 | 69.82 | 3,244 | 62.07 |  |
| female | 1,853 | 30.18 | 1,982 | 37.93 |  |
| Mean age STD | 44.58 | 17.71 | 36.86 | 11.76 |  |
| Homelessness n (%) |  |  |  |  | <0.001 |
| yes | 591 | 9.63 | 1958 | 37.47 |  |
| no | 5549 | 90.37 | 3268 | 62.53 |  |
| Lives in northern Ontario n (%) |  |  |  |  | <0.001 |
| yes | 5648 | 91.99 | 3753 | 71.81 |  |
| no | 492 | 8.01 | 1473 | 28.19 |  |
| Mental Disorder n (%) |  |  |  |  | <0.001 |
| yes | 657 | 10.68 | 823 | 15.71 |  |
| no | 5490 | 89.31 | 4417 | 84.29 |  |
| yes | 292 | 4.75 | 714 | 13.61 |  |
| Cellulitis n (%) |  |  |  |  | <0.001 |
| yes | 32 | 0.52 | 274 | 5.24 |  |
| no | 6108 | 99.48 | 4952 | 94.76 |  |

*n = number

*STD = standard deviation

**Table 2. Health service use by exposure groups.**

| | Alcohol | | Opioids | | p value |
|---|---|---|---|---|---|
| n (%) | 6,140 | 45.61 | 5,240 | 45.98 | |
| **ED only** n (%) | | | | | <0.001 |
| yes | 2956 | 48.14 | 2304 | 43.97 | |
| no | 3191 | 51.97 | 2936 | 56.03 | |
| **Direct admission** n (%) | | | | | 0.46 |
| yes | 347 | 5.65 | 313 | 5.97 | |
| no | 5800 | 94.35 | 4927 | 94.03 | |
| **AMU** n (%) | | | | | <0.001 |
| yes | 292 | 4.75 | 714 | 13.61 | |
| no | 5855 | 95.25 | 4527 | 86.39 | |
| **AMCS** n (%) | | | | | <0.001 |
| yes | 687 | 11.18 | 1095 | 20.90 | |
| no | 5460 | 88.82 | 4145 | 79.10 | |
| **Triage level** n (%) | | | | | 0.002 |
| 1 | 290 | 5.01 | 181 | 3.69 | |
| 2 | 1529 | 26.41 | 1326 | 27.02 | |
| 3 | 3167 | 54.70 | 2749 | 56.01 | |
| 4 | 731 | 12.63 | 559 | 11.39 | |
| 5 | 73 | 1.26 | 93 | 1.89 | |
| **30 day ED revisit** n (%) | | | | | |
| yes | 1235 | 41.53 | 1455 | 40.88 | 0.6 |
| no | 1739 | 58.47 | 2104 | 59.12 | |
| **Length of stay,** mean (STD) | 10.04 | 19.39 | 12.16 | 22.76 | |
| **Ed presentations,** mean (STD) | 6.21 | 6.21 | 6.4 | 6.39 | |

*n= number

*STD = standard deviation

*ED = emergency department

*AMU = addiction medicine unit

*AMCS = addiction consult service

comparison, 3% of individuals presenting with alcohol use were admitted the next day, and 8% were admitted within three days. Additionally, 30% of individuals presenting with opioid use were admitted to the hospital upon their next ED visit, compared to 50% of individuals presenting with alcohol use.

However, it was observed that the opioid group's likelihood of inpatient admission, as it related to the number of ED visits leading up to hospital admission, was significantly higher than that of the alcohol group. Table 3 shows that each additional ED visit is associated with a 28% increase in the hazard of inpatient admission for individuals with opioid-related issues compared to those with alcohol-related issues (Table 4). Specifically, the opioid group had a significantly higher likelihood of being admitted to the hospital as the number of ED visits increased. Fig 2 illustrates the cumulative incidence of hospital admission by exposure group based on the number of ED episodes leading to hospital admission.

## Discussion

In our study, we investigated the association between two types of substance use presentations in the ED and the subsequent risk of hospital admission. Our research yielded several key findings. We found high rates of next-day (14%)

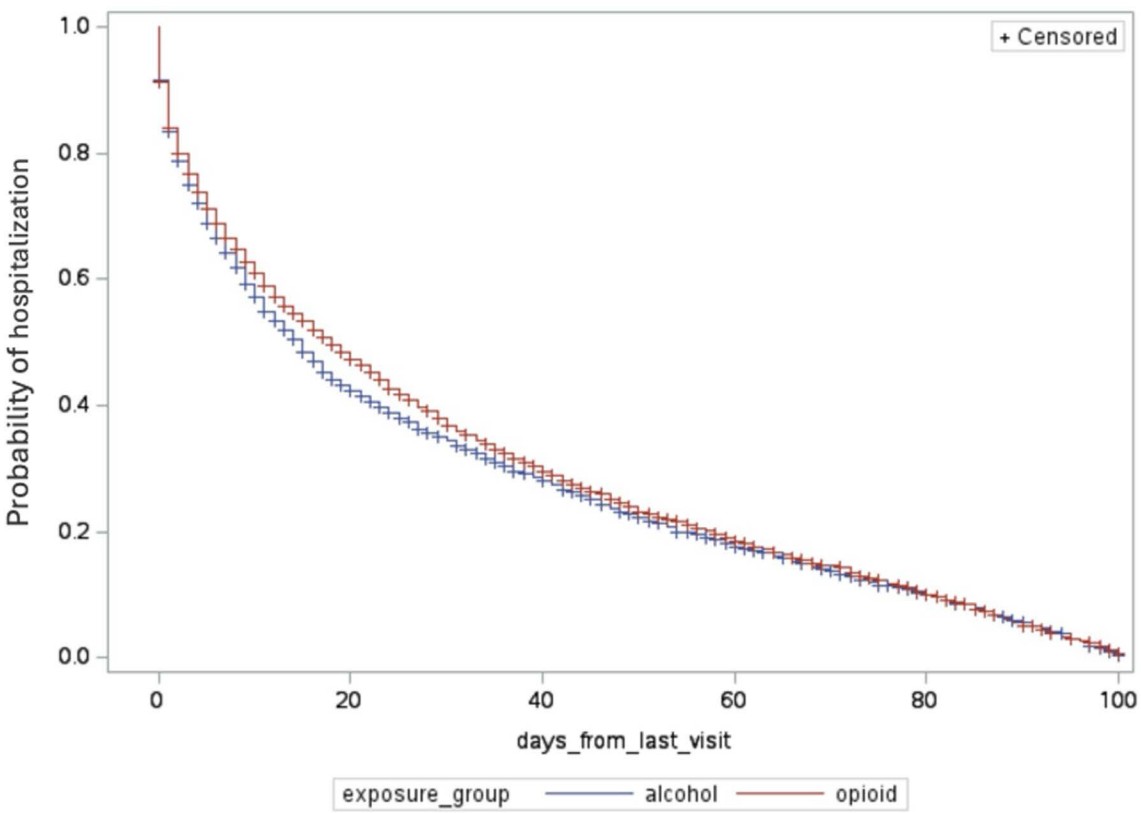

**Fig 1. Cumulative incidence of hospital admission by intervention in days.**

and 3-day (10%) hospital admissions after discharge from the ED among individuals presenting with opioid use. We observed lower rates of next-day (3%) and 3-day (8%) hospital admissions among those presenting with alcohol use. We also observed that 30% of individuals presenting with opioid use were admitted to the hospital upon their next ED visit, compared to 50% of individuals presenting with alcohol use. Additionally, each additional ED visit increases the risk of inpatient admission by 28% more for individuals with opioid-related issues compared to those with alcohol-related issues. Furthermore, 40.88% of individuals presenting with opioid use and 41.53% of those presenting with alcohol use revisit the ED within 30 days. We identified that experiencing homelessness is associated with a 40% increased likelihood of a 30-day revisit, and notably, a higher proportion of individuals presenting with opioids experience homelessness compared to those presenting with alcohol (37.47% versus 9.63%).

Our findings show heterogeneity among patients in the opioid and alcohol group presenting to HSN ED with regards to age, experience of homelessness, and infections. However, triage levels and healthcare utilization metrics such as length of stay and 30-day ED revisits were not significantly different. Our findings align with previous research indicating that patients with Opioid Use Disorder (OUD) are typically aged 35–45 years, with older patients seen in the ED visits group [19,20]. The higher rates of infection and homelessness are also supported in the literature [21, 22, 23].

The high rates of next-day and subsequent ED visit admissions for people presenting with opioid use and alcohol use indicate significant healthcare needs among these populations and possibly unmet needs based on lack of provider train-ing comfort with substance use disorders, and stigma. It has been shown that people who use substances are more likely

**Table 3. Factors associated with 30-day revisits (adjusted Odds Ratios and 95% confidence intervals).**

| Variable | Number of Revisit events | Adjuster Odds Ratio | 95% CI |
|---|---|---|---|
| **Sex** | | | |
| male | | | |
| female | | ref | |
| **Homelessness** | | 1.12 | (0.99 to 1.25) |
| yes | 1042 (23.9%) | 1.4* | (1.20 to 1.65) |
| no | 1009 (35%) | | |
| **ED only** | | | |
| yes | 1501 (15.88%) | 0.59 | (0.27 to 1.31) |
| no | 1839 (42.3%) | ref | |
| **Mental Disorder (s)** | | | |
| yes | 420 (14.52%) | 0.8* | (0.69 to 0.93) |
| no | 838 (19.26%) | ref | |
| **AMU** | | | |
| yes | 287 (9.6%) | 1.203 | (1.0 to 1.45) |
| no | 463 (10.64) | ref | |
| **AMCS** | | | |
| yes | 465 (16.1) | 1.09 | (0.94 to 1.27) |
| no | 804 (18.5) | ref | |
| **Cellulitis** | | | |
| yes | 79 (2.8) | 0.85 | (0.63 to 1.15) |
| no | 148 (3.4) | ref | |

*n= number

*STD = standard deviation

*ED = emergency department

*AMU = addiction medicine unit

*AMCS = addiction consult service

*ref = reference group for for regression model

**Table 4. Hazard ratio of hospital admission alcohol compared to opioid groups.**

| Variable | HR | (95% CI) | Adjusted HR | (95% CI) |
|---|---|---|---|---|
| Days from first ED visit to inpatient admission | 1.07 | (1.02 to 1.13) | 0.96 | (0.91 to 1.03) |
| Number of ED visits to Frst inpatient admission | 0.88 | (0.85 to 0.92) | *1.28 | (1.19 to 1.37) |

*HR =. Hazzard Ratio

*CI = confidence interval

to leave the ED without being seen, leave against medical advice (AMA), or be discharged prematurely, leading to acute subsequent healthcare events severe enough to necessitate hospitalization shortly after their ED discharge [24]. Premature discharges and leaving AMA have been associated with poor healthcare experiences, stigma, discrimination, and the underestimation of severe health events among people who use substances [25].

The findings that each additional ED visit increases the risk of inpatient admission by 28%for individuals with opioid-related issues compared to alcohol-related issues highlight the distinct healthcare challenges posed by these substances. Individuals in the opioid group required more ED visits before admission, suggesting more complex or severe health needs. While opioid use is often associated with acute crises necessitating immediate hospitalization, alcohol-related

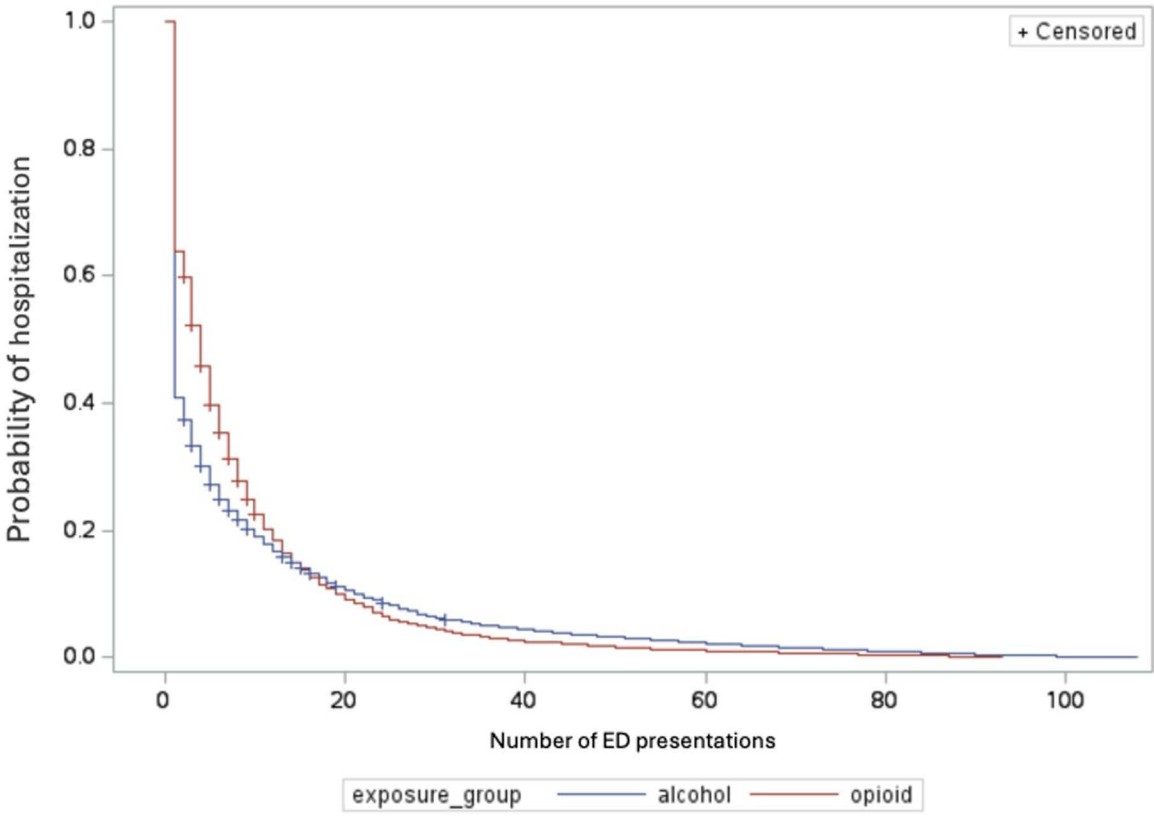

**Fig 2. Cumulative incidence of hospital admission by ED visits.**

admissions may reflect chronic conditions with differing trajectories of care. This distinction is an opportunity to improve healthcare interactions for people who use substances and better intervene at the point of contact to prevent suffering and reduce healthcare costs. Hospital admissions are often driven by withdrawal symptoms from alcohol and opioids, exacerbation of chronic conditions such as liver disease, or injuries and infections related to drug use [26, 27, 28]. Prompt initiation of evidence-based treatments, such as medication-assisted therapy can mitigate acute complications and reduce the need for repeat ED visits and costly hospital admission [29,30]. Additionally, implementing integrated care models that combine addiction treatment with emergency care such as addiction consult services can improve continuity of care, reduce unnecessary hospital admissions, increase connections with outpatient addiction supports, and address both acute and chronic health needs of individuals with substance use disorders [31, 32, 33].

To further support long-term recovery and reduce hospital utilization, initiating opioid management during hospitalization and ensuring coordinated transitions to outpatient care is essential. Structured discharge planning that includes OAT and follow-up care has been shown to be effective [28]. For example, the NavSTAR trial demonstrated that patient navigation, a model offering care coordination, motivational support, and linkage to community services which significantly reduced readmissions among individuals with comorbid substance use disorders [28]. Embedding such strategies into standard practice can improve treatment engagement and foster sustained recovery.

The study has noteworthy limitations. First, it relied on retrospective administrative data lacking clinical variables from patient charts, which limits the ability to establish causality and only allows for associations between exposure groups and outcomes. Covariates such as homelessness or visit reasons were operationalized, which may be prone to measurement

errors, and unobserved variables could potentially influence the associations between exposure groups and outcomes. Baseline variables are not static over time, and the study's findings may primarily apply to similar hospital settings, thus limiting generalizability. Additionally, events occurring outside of HSN, such as deaths or admissions to other hospitals, were not captured, although the nearest acute care facility is approximately 150 km away. Other limitations include the lack of generalizability to other care settings including those without addiction medicine consult services and addiction medicine units.

Selection bias may also be present, as individuals who seek care at the ED may differ in important ways from those who avoid or delay care. Furthermore, diagnostic coding may not fully capture the clinical complexity or context of substance-related presentations, introducing potential misclassification bias. These factors may have influenced the observed associations between ED presentation type and hospitalization outcomes, and should be considered when interpreting the findings.

## Conclusion

In conclusion, our findings underscore the differing healthcare needs and challenges among individuals presenting to the ED with opioid use versus alcohol use. Despite similar 30-day revisit rates, the patterns of hospital admissions, the influence of additional ED visits, and the higher prevalence of homelessness among people presenting with opioid use highlight the unique and acute healthcare vulnerabilities of this population. Opioid-related presentations were more frequently associated with complex or acute health issues requiring hospitalization, likely reflecting the urgent and severe nature of complications linked to opioid use. In contrast, alcohol-related presentations may reflect more chronic patterns of care need.

These distinctions emphasize the importance of tailoring interventions to address the specific needs of these populations. For individuals with opioid use, strategies such as prompt initiation of medication-assisted therapy, addressing infections, and implementing integrated care models are crucial. For those presenting with alcohol use, managing withdrawal symptoms and chronic conditions like liver disease remains vital. Across both groups, addressing stigma, improving provider training, and enhancing continuity of care can reduce premature discharges and promote more equitable and effective care delivery.

The observed association between homelessness and higher 30-day ED revisit rates underscores the need for focused interventions targeting this key social determinant of health. Integrating housing supports with medical care such as hospital-based housing referral programs, medical respite care, and partnerships with community housing organizations can improve stability and reduce repeated acute care use. Tailoring discharge planning to include connections to housing services and harm reduction supports may also enhance continuity of care and reduce the cycle of ED utilization among unhoused individuals.

## Acknowledgments

We want to express our gratitude to everyone who contributed to the study. Our most profound appreciation goes to the patients and staff who have and continue to help define research priorities and interpret results. We acknowledge the support of HSN leadership. We want to acknowledge Neil St Jean who provided line edits for this manuscript.

## Author contributions

**Conceptualization:** Kristen A. Morin, Laura Hill, Shannon Knowlan, Adele Bodson, Paola Nikodem, Natalie Aubin, David C Marsh, Tara Leary.

**Data curation:** Kristen A. Morin, Tara Leary.

**Formal analysis:** Kristen A. Morin.

**Funding acquisition:** Kristen A. Morin, Tara Leary.

**Investigation:** Kristen A. Morin, Tara Leary.

**Methodology:** Kristen A. Morin.

**Project administration:** Kristen A. Morin.

**Resources:** Kristen A. Morin.

**Supervision:** Kristen A. Morin, Shannon Knowlan, Natalie Aubin, David C Marsh.

**Validation:** Laura Hill, Shannon Knowlan, Adele Bodson, David C Marsh, Tara Leary.

**Visualization:** Kristen A. Morin, Paola Nikodem.

**Writing – original draft:** Kristen A. Morin, Adele Bodson, Paola Nikodem.

**Writing – review & editing:** Laura Hill, Shannon Knowlan, Adele Bodson, Paola Nikodem, Natalie Aubin, David C Marsh, Tara Leary.

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
