## [Decision Letter · Decision Letter 0]

16 Apr 2025

PONE-D-25-11470A retrospective cohort study evaluating the association between opioid and alcohol-related emergency department presentations and the subsequent risk of hospitalizationPLOS ONE

Dear Dr. Morin,

Thank you for submitting your manuscript to PLOS ONE. After careful consideration, we feel that it has merit but does not fully meet PLOS ONE’s publication criteria as it currently stands. Therefore, we invite you to submit a revised version of the manuscript that addresses the points raised during the review process. 

This study highlights a very important problem in the current medical scenario. There is a big disparity in the long-term management of patients with OUD and a high risk of increased healthcare utilization. This is a well-conducted study. Are there similar studies that looked into these two conditions and hospital admissions? If so, can you provide a background in the introduction section? Please refer to the questions brought on by the reviewers. 

Can you comment on strategies to engage patients for long-term outpatient care - starting opioid management at the time of hospital discharge and linkage with outpatient primary care and opioid management?  One of the studies that looked into this is the NavSTAR trial which showed that patient navigation could reduce hospital admissions amongst patients with comorbid SUDs. 

(https://doi.org/10.1186/s13722-024-00463-9 ) 

The study has identified some key findings including social determinants that may affect the readmissions in ED including homelessness. Strategies to address this issue can be included in the conclusion section. 

We look forward to receiving your revised manuscript.

Kind regards,

Arunima Dutta, MD, FACP, FAPCR

Academic Editor

PLOS ONE

Additional Editor Comments (if provided):

Reviewers' comments:

Reviewer's Responses to Questions

**Comments to the Author**

1. Is the manuscript technically sound, and do the data support the conclusions?

Reviewer #1: Yes

Reviewer #2: Yes

Reviewer #3: Yes

2. Has the statistical analysis been performed appropriately and rigorously? 

Reviewer #1: Yes

Reviewer #2: Yes

Reviewer #3: Yes

3. Have the authors made all data underlying the findings in their manuscript fully available?

Reviewer #1: Yes

Reviewer #2: Yes

Reviewer #3: Yes

4. Is the manuscript presented in an intelligible fashion and written in standard English?

Reviewer #1: Yes

Reviewer #2: Yes

Reviewer #3: Yes

5. Review Comments to the Author

Reviewer #1: Thank you for this important and timely study.

I would suggest that utilizing more recent references would strengthen your manuscript as many in the introduction are very old, especially 1, 2, 5, 6, 24. More current relevant references on opioids can be found in the 2024 CRISM guideline for managing opioid use disorder (CMAJ 2024 November 12;196:E1280-90. doi: 10.1503/cmaj.241173) and CCSA reports.

The first description triage as a variable is in the results section and I suggest providing some introduction to this variable earlier on.

Please also pay more attention to formatting of the references and manuscript text.

I wonder if including the 320 patients with both opioid and alcohol presentations in the opioid group has skewed the results and suggest excluding those patients or at least running subgroup analyses without them.

Reviewer #2: The authors present a compelling argument for examining the association between opioid use disorder (OUD) and alcohol use disorder presentations in the emergency department (ED) and the subsequent risk of hospital admission. This study provides valuable insights into the differing impacts of these substance use disorders on healthcare utilization. The methodology is sound, and the manuscript is well-organized. The conclusion—that individuals with OUD have a higher likelihood of inpatient admission and ED visits—emphasizes the need for targeted interventions and resources for this high-risk population. The authors appropriately acknowledge the study limitations, including challenges related to operationalizing measured constructs and the limited generalizability of findings. However, the method section requires further development. Additional detail is needed regarding the data collection and de-identification processes. How were the data obtained or extracted? Are these electronic medical record data? What specific measures were used to ensure participant anonymity (de-identification process)? Furthermore, the research questions and hypotheses are not clearly articulated. It is essential to explain why particular covariates were selected and how they were theoretically or empirically justified. Were these based on prior literature or pre-specified hypotheses? Clarifying which covariates were controlled for in the logistic regression analysis would also strengthen the rigor and transparency of the analytic approach.

Recommendations:

- Provide a clear and concise statement of your research questions and hypotheses in the introduction.

- Elaborate on your data collection and data de-identification process in the method section.

- Ensure your research questions and hypotheses are consistently addressed in the data analysis plan and results sections.

- Finally, clarify the rationale for covariate selection and specify which were included or controlled for in the regression models.

Overall Evaluation:

This manuscript offers important contributions to understanding the healthcare impact of OUD compared to alcohol use disorder. By strengthening the articulation of the study's research questions, hypotheses, data analysis plan, and the rationale behind methodological choices, the authors can improve the clarity, focus, and overall impact of the work.

Reviewer #3: Dear author, despite of the study design and limitations explained in the article. This thematic is absolutely necessary to understand this population and their needs. I have no suggestion to methodology however you have to identify the bias and how they have impacted in the results.

6. PLOS authors have the option to publish the peer review history of their article (what does this mean? ). If published, this will include your full peer review and any attached files.

**Do you want your identity to be public for this peer review?** For information about this choice, including consent withdrawal, please see our Privacy Policy .

Reviewer #1: No

Reviewer #2: No

Reviewer #3: No

---

## [Author Response · Author response to Decision Letter 1]

24 Apr 2025

Response to reviewers – Round 1

A retrospective cohort study evaluating the association between opioid and alcohol-related emergency department presentations and the subsequent risk of hospitalization

Editors Comments:

This study highlights a very important problem in the current medical scenario. There is a big disparity in the long-term management of patients with OUD and a high risk of increased healthcare utilization. This is a well-conducted study. Are there similar studies that looked into these two conditions and hospital admissions? If so, can you provide a background in the introduction section? Please refer to the questions brought on by the reviewers.

• Thank you for your comment, previous studies have been highlighted in the introduction starting line 94.

Can you comment on strategies to engage patients for long-term outpatient care - starting opioid management at the time of hospital discharge and linkage with outpatient primary care and opioid management? One of the studies that looked into this is the NavSTAR trial which showed that patient navigation could reduce hospital admissions amongst patients with comorbid SUDs.

(https://doi.org/10.1186/s13722-024-00463-9)

• Thank you for your suggestion, please see the added section starting line 327.

The study has identified some key findings including social determinants that may affect the readmissions in ED including homelessness. Strategies to address this issue can be included in the conclusion section.

• Thank you for your comment, please see revision starting line 375.

Reviewer #1

Thank you for this important and timely study.

I would suggest that utilizing more recent references would strengthen your manuscript as many in the introduction are very old, especially 1, 2, 5, 6, 24. More current relevant references on opioids can be found in the 2024 CRISM guideline for managing opioid use disorder (CMAJ 2024 November 12;196:E1280-90. doi: 10.1503/cmaj.241173) and CCSA reports.

• Thank you for your suggestion, more up to date references have been added in the introduction and throughout the paper to strengthen the overall manuscript.

The first description triage as a variable is in the results section and I suggest providing some introduction to this variable earlier on.

• Thank you, a section introducing triage as a variable has been added on line 163.

Please also pay more attention to formatting of the references and manuscript text.

• Thank you, referencing has been standardized throughout the manuscript using Vancouver style referencing.

I wonder if including the 320 patients with both opioid and alcohol presentations in the opioid group has skewed the results and suggest excluding those patients or at least running subgroup analyses without them.

• Thank you. We chose to default patients to the opioid group and adjust for alcohol use because it opioid use typically poses higher immediate risks, such as overdose, and is the primary focus of treatment with medications like opioid agonist therapy. This classification reflects the clinical emphasis on managing opioid use disorder, especially when opioid-related complications are more pronounced than alcohol use.

Reviewer #2

Recommendations:

- Provide a clear and concise statement of your research questions and hypotheses in the introduction.

• Thank you for your recommendation, please see an addition to the introduction clearly highlighting our research statement and hypotheses.

- Elaborate on your data collection and data de-identification process in the method section.

• Thank you, this was added on line 123.

- Ensure your research questions and hypotheses are consistently addressed in the data analysis plan and results sections.

• Thank you for your recommendation, added sections starting line 186 and line 191 have been added.

-Finally, clarify the rationale for covariate selection and specify which were included or controlled for in the regression models.

• Thank you this was added on line 175.

Overall Evaluation:

This manuscript offers important contributions to understanding the healthcare impact of OUD compared to alcohol use disorder. By strengthening the articulation of the study's research questions, hypotheses, data analysis plan, and the rationale behind methodological choices, the authors can improve the clarity, focus, and overall impact of the work.

• We sincerely appreciate your detailed and insightful review.

Reviewer #3: Dear author, despite of the study design and limitations explained in the article. This thematic is absolutely necessary to understand this population and their needs. I have no suggestion to methodology however you have to identify the bias and how they have impacted in the results.

• Thank you for your comment, a section has been added on line 348 to identify the potential bias, and their impact on results.

---

## [Editor Report · Decision Letter 1]

6 May 2025

A retrospective cohort study evaluating the association between opioid and alcohol-related emergency department presentations and the subsequent risk of hospitalization

PONE-D-25-11470R1

Dear Dr. Dr Morin,

We’re pleased to inform you that your revised manuscript has been judged scientifically suitable for publication and will be formally accepted for publication once it meets all outstanding technical requirements.

Kind regards,

Arunima Dutta, MD, FACP, FAPCR

Academic Editor

PLOS ONE

---

## [Editor Report · Acceptance letter]

PONE-D-25-11470R1

PLOS ONE

Dear Dr. Morin,

I'm pleased to inform you that your manuscript has been deemed suitable for publication in PLOS ONE. Congratulations! Your manuscript is now being handed over to our production team.

Kind regards,

on behalf of

Dr. Arunima Dutta

Academic Editor

PLOS ONE